# In vivo transplantation of 3D encapsulated ovarian constructs in rats corrects abnormalities of ovarian failure

Sivanandane Sittadjody[1], Justin M. Saul[2], John P. McQuilling[1,3], Sunyoung Joo[1], Thomas C. Register [4], James J. Yoo[1,3], Anthony Atala[1,3] & Emmanuel C. Opara [1,3]

Safe clinical hormone replacement (HR) will likely become increasingly important in the growing populations of aged women and cancer patients undergoing treatments that ablate the ovaries. Cell-based HRT (cHRT) is an alternative approach that may allow certain physiological outcomes to be achieved with lower circulating hormone levels than pharmacological means due to participation of cells in the hypothalamus-pituitary-ovary feedback control loop. Here we describe the in vivo performance of 3D bioengineered ovarian constructs that recapitulate native cell–cell interactions between ovarian granulosa and theca cells as an approach to cHRT. The constructs are fabricated using either $Ca^{++}$ or $Sr^{++}$ to crosslink alginate. Following implantation in ovariectomized (ovx) rats, the $Sr^{++}$-cross-linked constructs achieve stable secretion of hormones during 90 days of study. Further, we show these constructs with isogeneic cells to be effective in ameliorating adverse effects of hormone deficiency, including bone health, uterine health, and body composition in this rat model.

[1] Wake Forest Institute for Regenerative Medicine, Wake Forest School for Medicine, Winston-Salem, NC 27157, USA. [2] Department of Chemical, Paper and Biomedical Engineering, Miami University, Oxford, OH 45056, USA. [3] School of Biomedical Engineering and Sciences (SBES), Wake Forest School of Medicine, Winston-Salem, NC 27157, USA. [4] Department of Pathology, Section on Comparative Medicine, Wake Forest University School of Medicine, Winston-Salem, NC 27157, USA. Correspondence and requests for materials should be addressed to E.C.O. (email: eopara@wakehealth.edu)

ancer therapies and the growing number of women reaching the age of menopause have led to an increasing prevalence in the loss of ovarian function, which has profound health complications including sexual disturbances, obesity, and osteoporosis[1, 2]. Pharmacologic hormone replacement therapy (pHRT) with estrogen alone or estrogen and progestogens is known to effectively ameliorate these effects, but these modes of pHRT are controversial and their use has decreased[3, 4]. The decrease in use of pHRT is primarily an outcome of the Women's Health Initiative (WHI) studies of 2002 and 2004, which indicated that adverse effects, including breast, endometrial, and ovarian cancers[5, 6] outweighed benefits such as reductions in osteoporotic fractures[7, 8]. However, pHRT may have beneficial effects[9–11], in particular when delivered at an optimal dosage, frequency, and appropriate time[12]. Thus, methods of HRT delivery that can maintain beneficial effects (e.g., enhanced bone mineral density) with improved safety profiles (e.g., no increased risk of cancer) are needed for the treatment of conditions associated with loss of ovarian function such as osteoporosis.

Unfortunately, achieving the optimal dosage delivery of HRT is challenging, owing to the complexity of the endocrine system. Granulosa and theca cells of the ovary produce estradiol ($E_2$) and progesterone ($P_4$) in response to follicle-stimulating hormone (FSH) and luteinizing hormone (LH) from the pituitary. The secretion of LH and FSH, in turn, is regulated by gonadotropin-releasing hormone (GnRH) produced by the hypothalamus. Further, the hormones of the ovary ($E_2$, $P_4$, activin, and inhibin) provide feedback to the hypothalamus and pituitary, thereby regulating their own production in the hypothalamic-pituitary-ovarian (HPO) endocrine axis[13, 14]. Pharmacological approaches to HRT lack integration into the hypothalamic-pituitary (HP) components of the axis that would allow feedback and regulation over dosage and timing of circulating hormone levels associated with the delivery method. As such, pHRT methods exhibit different plasma concentrations of hormones from those associated with functional ovaries, which may contribute to safety issues associated with pHRT.

Regenerative medicine approaches that use cell-based hormone replacement therapy (cHRT) offer a potential solution to temporal control of hormone delivery and the ability to restore the HPO axis in a way not possible with pHRT. We hypothesized that by engineering a cell encapsulation process to more faithfully recapitulate native ovarian structure, the key functional effects of circulating hormones (which are sensitive to dosage and time) could be achieved more effectively and safely than pHRT. We have previously described[15] an approach to achieve microencapsulation of ovarian cells that results in bioengineered constructs that replicate key structure-function relationships of ovarian follicles (Fig. 1a), as an approach to cHRT. In this report, we have adapted an isogeneic cell-based construct to provide a proof-of-concept for the potential benefits of cHRT.

## Results

**Construct preparation and evaluation.** The fabricated constructs (shown schematically in Fig. 1b) have distinct compartments for the granulosa and theca cells as indicated by confocal microscopy of the constructs containing specifically labeled cells (Fig. 1c). For all in vivo studies, only isogeneic ovarian cell-based constructs were used to demonstrate proof-of-concept, and the constructs were implanted into omental pouches created in ovx rats. Retrieval of the constructs at 90 days post-implantation showed that the constructs remained in the omental pouch and that the omental region around the capsules was well-vascularized (Fig. 1d). In addition, we did not observe any evidence of

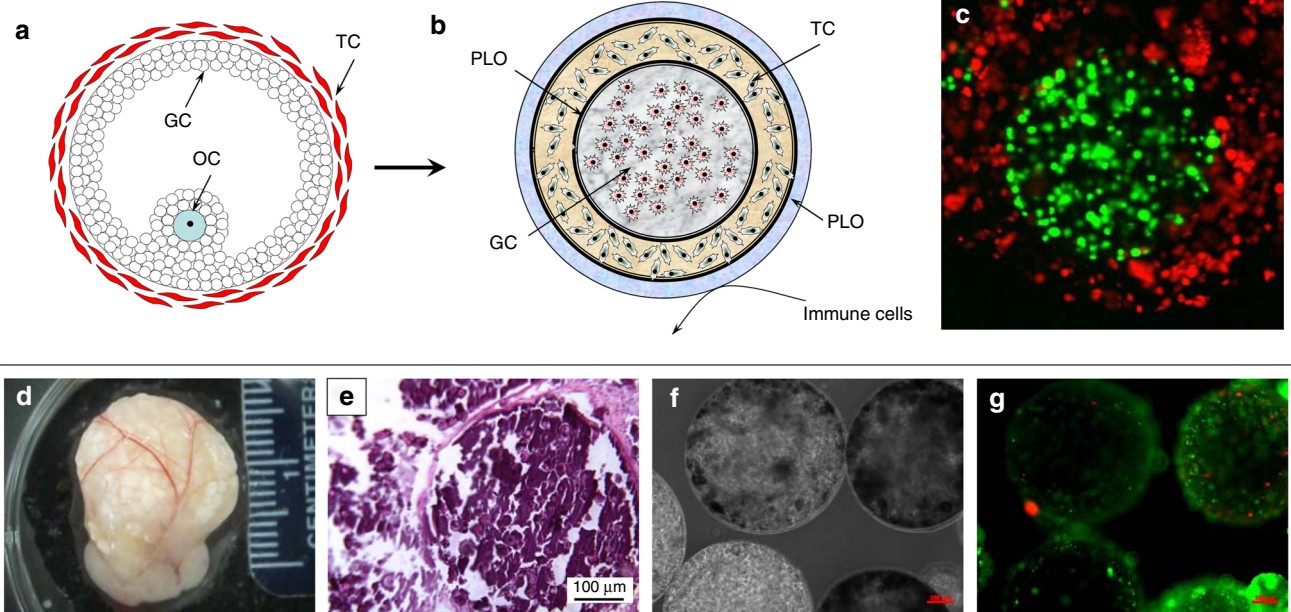

**Fig. 1** Ovarian construct fabrication and characterization and explants from in vivo studies. Schematic diagram of a native ovarian follicle (**a**) compared to the bioengineered ovarian construct (**b**). 3D-confocal images of bioengineered ovarian construct (**c**) demonstrating compartmentalization of different cells within the constructs as determined through the use of CellTracker green-labeled cells (granulosa) in the inner layer and CellTracker orange-labeled cells (theca) in the outer layer. Images of bioengineered ovarian construct retrieved 90 days after transplantation into ovariectomized rats including the presence of the vascularized omentum pouch enclosing the constructs following explantation (**d**). Explanted constructs showed minimal fibrous encapsulation as indicated by H&E staining (**e**). Phase-contrast images of the microcapsules after retrieval show that the constructs remain intact throughout the 90-day period tested in vivo (**f**). Live/dead imaging of the retrieved capsules (**g**), where green indicates live and red indicates dead cells, which shows that most cells in the constructs remained viable during the 90-day implantation period. Scale bars are 100 μm for **e–g**

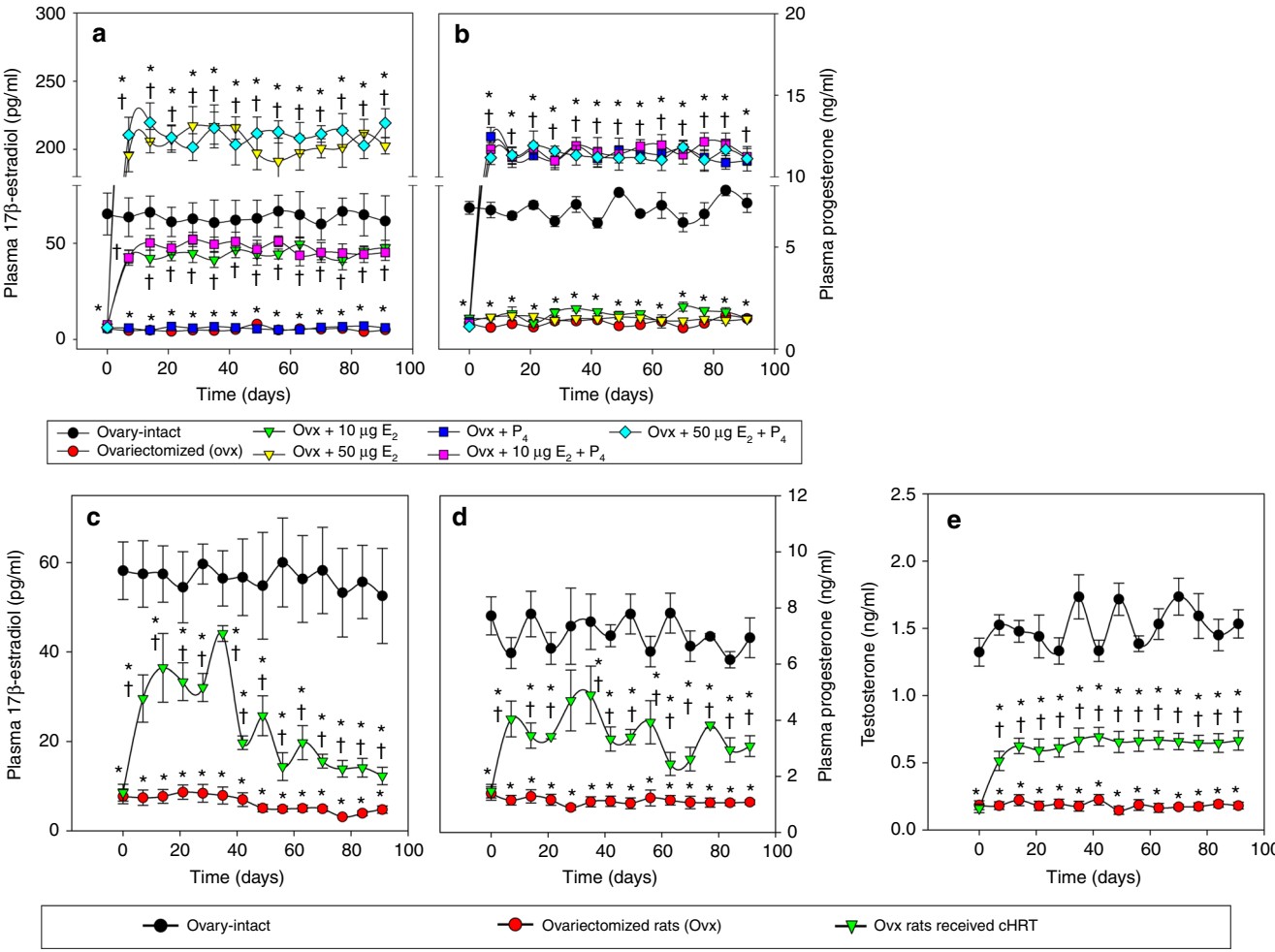

**Fig. 2** Plasma steroid hormone levels in sham and ovx rats with or without treatment. Circulating plasma levels of 17β-estradiol ($E_2$; **a**, **c**) or progesterone ($P_4$; **b**, **d**) for ovariectomized (ovx) rats transplanted with hormone pellets for the pharmacological-HRT (pHRT) treatments (**a**, **b**) or bioengineered ovarian constructs (**c**, **d**) for the cell-based-HRT (cHRT) treatment and testosterone (**e**) for ovx rats treated with bioengineered ovarian constructs. For pHRT **a**, **b**, 0.25 mg $E_2$ pellets with a release kinetics of 10 µg/kg per day were implanted to achieve low-dose $E_2$ delivery, while 1.25 mg $E_2$ pellets with a release kinetics of 50 µg/kg per day were implanted to achieve high-dose $E_2$ delivery with or without a fixed dose of 50 mg $P_4$ pellets with a release kinetics of 2 mg/kg per day to achieve a physiological dose of $P_4$. The figures represent the data from one of three separate sets of experiments ($n = 10$ for each experiment). Each data point represents the mean ± SEM of 10 animals (biological replicates) with each data point as a single technical experiment. * indicates significance $p < 0.05$ compared to ovary-intact rats; † indicates significance $p < 0.05$ compared to ovx rats as determined by a one-way ANOVA followed by Student–Newman–Keuls post hoc analysis

fibrous encapsulation by either gross inspection (Fig. 1d) or histology (Fig. 1e). The cells remained encapsulated as indicated by phase images of the retrieved constructs (Fig. 1f) and also maintained high levels of viability (Fig. 1g) over the time course of the studies. Although we did not assess the degree of degradation in the constructs, we believe that these observations are consistent with the stability of the construct over the period of experiment.

**Plasma levels of hormones**. A potential advantage of a cHRT system is the ability of the construct to participate in the HPO endocrine loop for regulation of plasma hormone concentrations. In the first set of experiments with constructs in which alginate was cross-linked with calcium ions ($Ca^{++}$), we therefore assessed circulating levels of 17β-estradiol ($E_2$), progesterone ($P_4$) (Fig. 2), and gonadotropins (FSH and LH) (Fig. 3) after pHRT (implanted subcutaneously) or cHRT implantations in omentum pouches created in ovx rats. Untreated ovx rats (ovx with placebo pellets not loaded with hormones in the case of pHRT, or animals

receiving empty alginate constructs and no cells in the case of cHRT) and sham-operated (ovary-intact) rats served as controls. To further investigate the ability of cells in the cHRT constructs to secrete androgen precursors for estrogen or to participate in the HPO axis, we also investigated plasma levels of testosterone (Fig. 2e) and inhibin (Fig. 3e), respectively.

The pHRT treatment delivered by a subcutaneous route was given at both low (10 µg/kg b.w.) and high (50 µg/kg b.w.) doses of $E_2$ either in the absence or presence of $P_4$. These doses were chosen based on previous studies[16–18]. Ovx rats that received placebo pellets were used as negative controls, while sham-operated ovary-intact rats served as positive controls. The pHRT regimens used resulted in various plasma levels of $E_2$ and $P_4$ depending on the dosage given (Fig. 2a, b). Low-dose estrogen formulations led to slightly elevated levels of $E_2$ compared to our bioengineered cHRT constructs (compare Fig. 2a with 2c), but were still less than the hormone levels measured in ovary-intact animals. The high-dose regimens led to supraphysiological levels of $E_2$ in the plasma, as indicated by values apparently above those measured in ovary-intact animals (Fig. 2a).

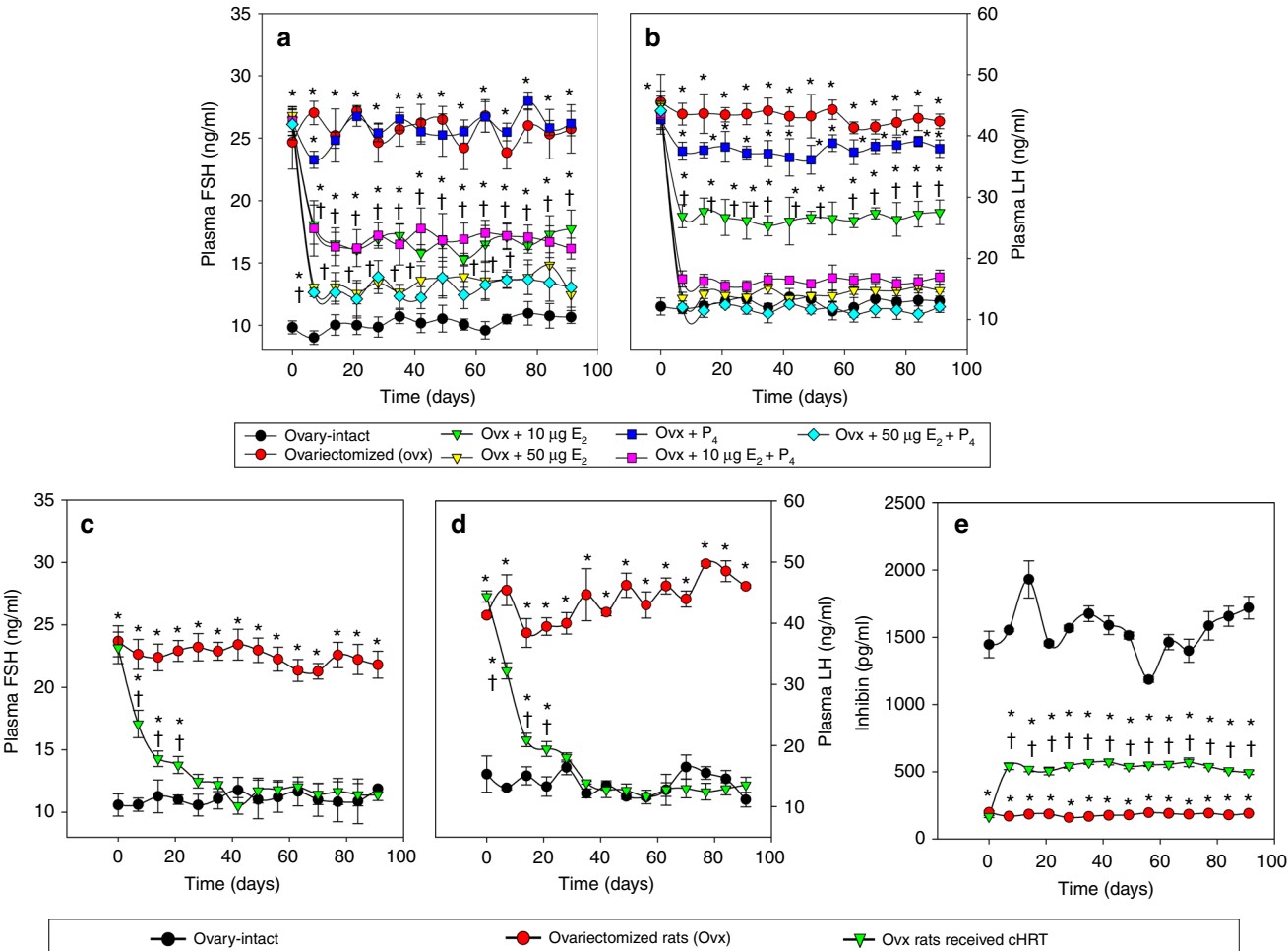

**Fig. 3** Effect of pHRT and cHRT treatments on the negative feedback loop in ovx rats. Circulating levels of FSH (**a**, **c**) and LH (**b**, **d**) in rats implanted with hormone pellets (**a**, **b**) and ovarian cell constructs (**c**, **d**), respectively. Circulating plasma levels of inhibin (**e**) for ovx rats treated with cHRT constructs. The figures represent data from one of three separate experiments ($n = 10$ for each experiment). Each data point represents the mean ± SEM of 10 animals (biological replicates) with each data point as a single technical experiment. * indicates significance $p < 0.05$ compared to ovary-intact rats; † indicates significance $p < 0.05$ compared to ovx rats as determined by ANOVA followed by a Student–Newman–Keuls post hoc analysis

$E_2$ and $P_4$ levels in animals treated with our bioengineered cHRT constructs did not reach the levels measured in ovary-intact rats. Nonetheless, plasma levels of $E_2$ and $P_4$ in ovx rats that received the bioengineered constructs were significantly higher than levels measured in untreated ovx rats throughout the study period (Fig. 2c, d; $p < 0.05$ at all-time points using one-way analysis of variance (ANOVA) followed by Student–Newman–Keuls post hoc analysis).

We also evaluated the plasma levels of FSH and LH following treatment of ovx animals with either pHRT or the bioengineered constructs for cHRT (Fig. 3). Plasma levels of both FSH and LH were found to be elevated in ovx rats due to the loss of negative feedback control loop. Although $P_4$ pellet implantation did not exert any impact on the FSH levels, the implantation of $E_2$ pellets in ovx rats showed a dose-dependent suppression of the elevated FSH. However, even the high-dose $E_2$ (50 μg/kg) + $P_4$ administration did not completely suppress the elevated FSH levels to the normal levels of ovary-intact animals (Fig. 3a). This is in contrast to the cHRT recipients (Fig. 3c), in which FSH levels were suppressed comparable to those of ovary-intact rats. The failure of the pHRT treatment to normalize FSH levels is likely due to absence of other regulators involved in the negative feedback loop from ovarian tissues such as inhibin. On the other hand, the high-

dose 50 μg/kg $E_2$ pellets with and without combination with $P_4$, as well as the low-dose 10 μg/kg $E_2$ plus $P_4$ schemes brought the plasma levels of LH down to the level measured in ovary-intact rats (Fig. 3b). However, there was only a slight decline in the circulating levels of LH in the group that received $P_4$ pellets alone (Fig. 3b). Likewise, the group of ovx rats implanted with 10 μg/kg $E_2$ alone showed an intermediate response in the suppression of LH. Even though the circulating levels of $E_2$ and $P_4$ in the ovx rats that received cHRT constructs were found to be mid-way between ovary-intact rats and ovx rats (Fig. 2), the full restoration of the negative feedback loop is evident from the suppression of both gonadotropins to the levels that are comparable to ovary-intact rats by 3 weeks after implantation (Fig. 3c, d).

As shown in Figs. 2e and 3e, the plasma levels of both testosterone (Fig. 2e) and inhibin (Fig. 3e) produced by the cHRT constructs were intermediate between those of the ovx and ovary-intact rats. The testosterone levels are consistent with the levels of production of estrogen and progesterone. In general, our data indicate that the cellular constructs are producing additional hormones (inhibin) involved in the HPO axis and androgens (testosterone) that serve as precursors to estrogen production, a physiologic phenomenon not achievable with pHRT.

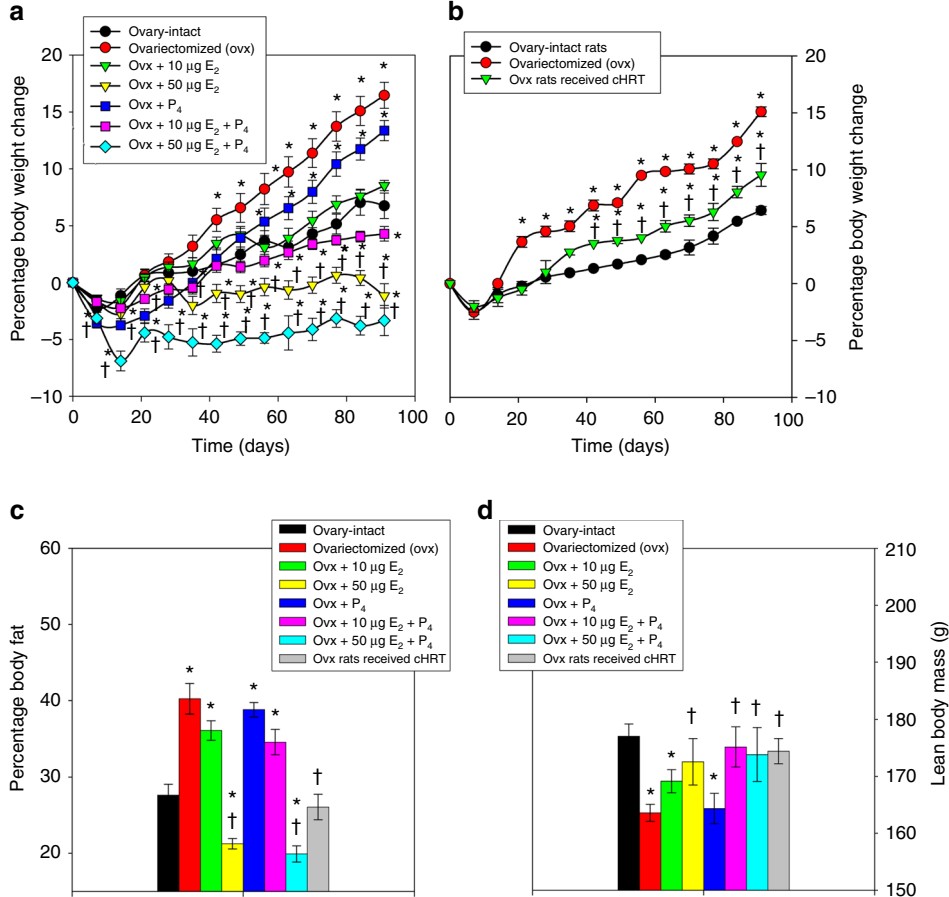

**Fig. 4** Body weight changes and fat accumulation in sham and ovx rats with or without treatment. Changes in body weight (**a**, **b**) and percentage fat accumulations (**c**) in ovx rats implanted with either hormone pellets (**a**, **c**) or ovarian constructs (**b**, **c**) compared to ovary-intact rats and ovx control groups. DXA scans for lean body mass with each formulation (**d**). Each data point represents the mean ± SEM of 10 animals (biological replicates) with each data point as a single technical experiment. * indicates significance $p < 0.05$ compared to ovary-intact rats; † indicates significance $p < 0.05$ compared to ovx rats as determined by one-way ANOVA followed by Student–Newman–Keuls post hoc analysis

**Body weight and fat accumulation**. It is well known that loss of ovarian function leads to body fat accumulation and body weight gain[19–21] as shown in Fig. 4 comparing ovary-intact and ovx rats. We evaluated the influence of hormone supplementation by the pHRT and cHRT constructs on body weight by weekly monitoring, while fat accumulation and lean body mass were measured at the end of the study by using dual-energy X-ray absorptiometry (DXA) imaging. Ovx rats receiving low-dose $E_2$ alone or with $P_4$ (Fig. 4a) and ovx rats treated with the cHRT bioengineered constructs (Fig. 4b) had body weights comparable to the levels in ovary-intact animals. Interestingly, the percent body fat observed with the low-dose pHRT regimens was higher than the amount observed with the cHRT bioengineered constructs (Fig. 4c). In other words, despite similar plasma levels of circulating hormones, the cHRT constructs were able to achieve substantially lower percent body fat levels than low-dose (10 μg $E_2 \pm P_4$) treatments. Furthermore, the body fat percentages in animals receiving the cHRT constructs were not different from ovary-intact animals (Fig. 4c). Consistent with observations in other studies[19, 22], the high-dose $E_2$ (supraphysiological) administration (with or without $P_4$) resulted in a greatly reduced body weight compared to ovary-intact animals (Fig. 4c). There was no difference between the effects of pHRT regimens (low dose or high dose) containing estrogen and the cHRT treatments in terms of lean body mass as determined by DXA (Fig. 4d), indicating that the observed effect of estrogen deprivation on body weight

was related to fat deposition. We also note that whole body BMD was substantially improved with our ovarian constructs compared to ovx rats (Supplementary Fig. 1A).

**Characterization of bone metabolism and architecture**. Bone is a major target organ of estrogens, and estrogen deficiency can lead to osteopenia, osteoporosis, and osteoporotic fractures in pre- and post-menopausal women. We therefore assessed the impact of our bioengineered ovarian cHRT constructs on circulating biomarkers of bone metabolism, trabecular bone microarchitecture, and bone mineral density (BMD). The systemic influence of sex steroid hormones secreted by the ovarian constructs on bone metabolism is demonstrated by the fact that plasma levels of osteocalcin (OCN) and c-telopeptide of collagen (CTx) were substantially reduced in animals treated with the ovarian constructs compared to ovx rats (Supplementary Fig. 1B, C), as OCN levels in cHRT construct recipients were similar to those measured in ovary-intact rats by 30 days (Supplementary Fig. 1B).

BMD was determined by micro-computed tomography (μCT) applied to the proximal femur at 90 days post-implantation. Figure 5a shows a quantification of the femoral BMD for control animals (ovary-intact or ovx) with comparison to the various pHRT treatments or to our cHRT bioengineered ovarian construct treatment. All treatment groups except $P_4$ pellet

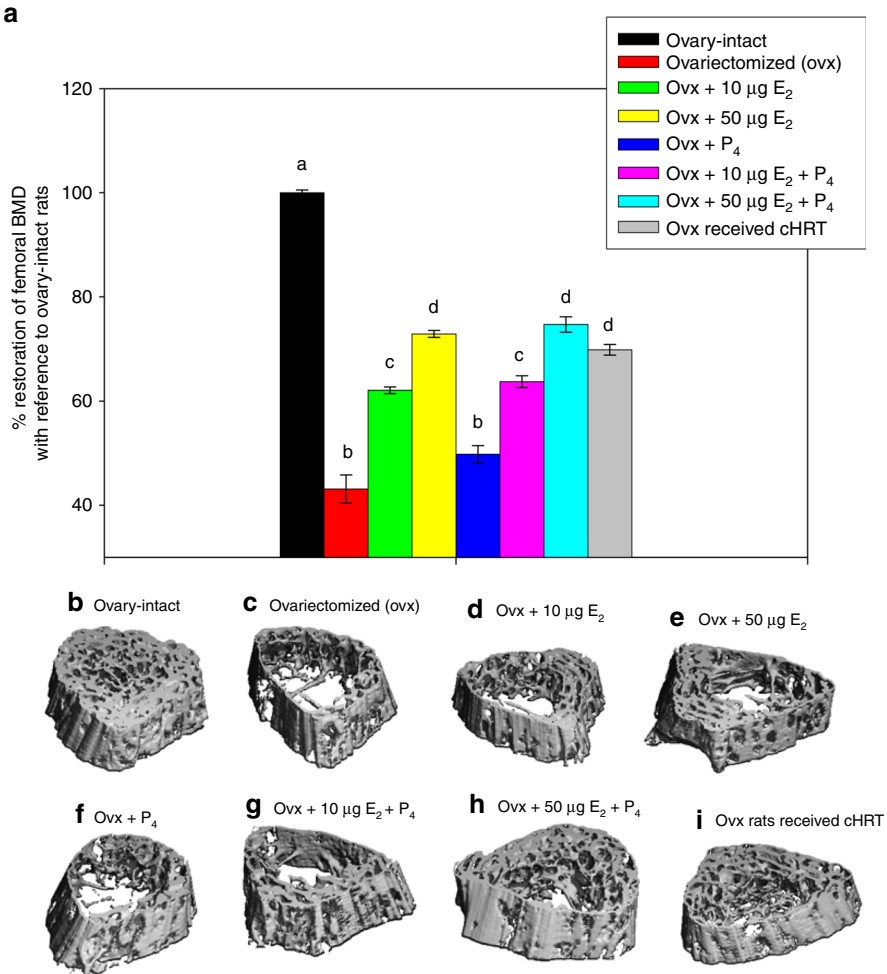

**Fig. 5** Bone mineral density and micro-architecture in sham and ovx rats with or without treatment. Percent of restoration of femoral BMD as determined quantitatively by μ-CT scans (**a**) or qualitatively by representative scans for the various treatments (**c–i**) and compared to ovary-intact animals (**b**). Each data point represents the mean ± SEM of 10 animals (biological replicates) with each data point as a single technical experiment. Different letters above the bar represent statistical significance at $p < 0.05$ between the groups. Same letters denote no significant difference between the groups as determined by one-way ANOVA followed by Student–Newman–Keuls post hoc analysis

implants alone showed substantial increases in BMD compared to ovx animals, but none of the treatment groups had the BMD restored to the level of ovary-intact animals. The cHRT bioengineered construct recipients achieved BMD levels that were not different from the high-dose $E_2$ (supraphysiological) but substantially greater than the levels attained with low-dose E2. This is particularly striking given the fact that the bioengineered constructs had lower circulating plasma levels of $E_2$ and $P_4$ than both the low-dose $E_2$ and the high-dose $E_2$ pHRT regimens.

The micro-CT scans shown in Fig. 5b–i indicate differences in the bone architecture and porosity between the various treatments. In particular, bone from animals treated with the bioengineered constructs (Fig. 5i) showed morphology similar to the native bone (ovary-intact animals; Fig. 5b), with the lowest levels of porosity across the bone and regions of both cortical and trabecular bone. Animals that received pHRT containing estrogen (Fig. 5d, e, g, h), and particularly those receiving the high-dose estrogen (Fig. 5e, h) showed thicker cortical bone than ovx rats, but there was also demonstrable porosity in the trabecular region. This observation may account for the measurements of increased BMD in the 3D images of the μCT due to the higher density and mineral content of cortical bone[23]. In short, although BMD levels were similar for cHRT and high-dose estrogen pHRT, BMD and porosity appear to be better for cHRT in the bone cross section.

Taken as a whole, the results shown in Fig. 5 indicate that the cHRT treatment led to better bone outcomes (BMD and bone porosity) than the pHRT treatments.

Although the reason for the reduced levels of porosity achieved with the bioengineered constructs is not clear, effects of other cell-based products besides estrogen and progesterone may play a role. Quantitative evaluation of trabecular fractional bone volume, trabecular number, trabecular thickness and trabecular separation also demonstrated preservation of femoral bone health in ovx animals treated with our bioengineered ovarian constructs (Supplementary Table 1).

**Changes in uterine morphometry**. The loss of ovarian function is known to have adverse effects on the genitourinary tract in women, which can lead to sexual dysfunction, urinary incontinence, and perturbations in the structure of sex organs[24]. The uterus is one target of ovarian steroids and is very sensitive to the levels of estrogens and progestins to which it is exposed. Indeed, studies have shown that uterine tissue is susceptible to hyperplasia and hypertrophy when treated with estrogens[25, 26].

Therefore, we evaluated morphometrical changes in uterine tissue at the end of the study to determine the effects of both pHRT and our cHRT bioengineered construct treatments on

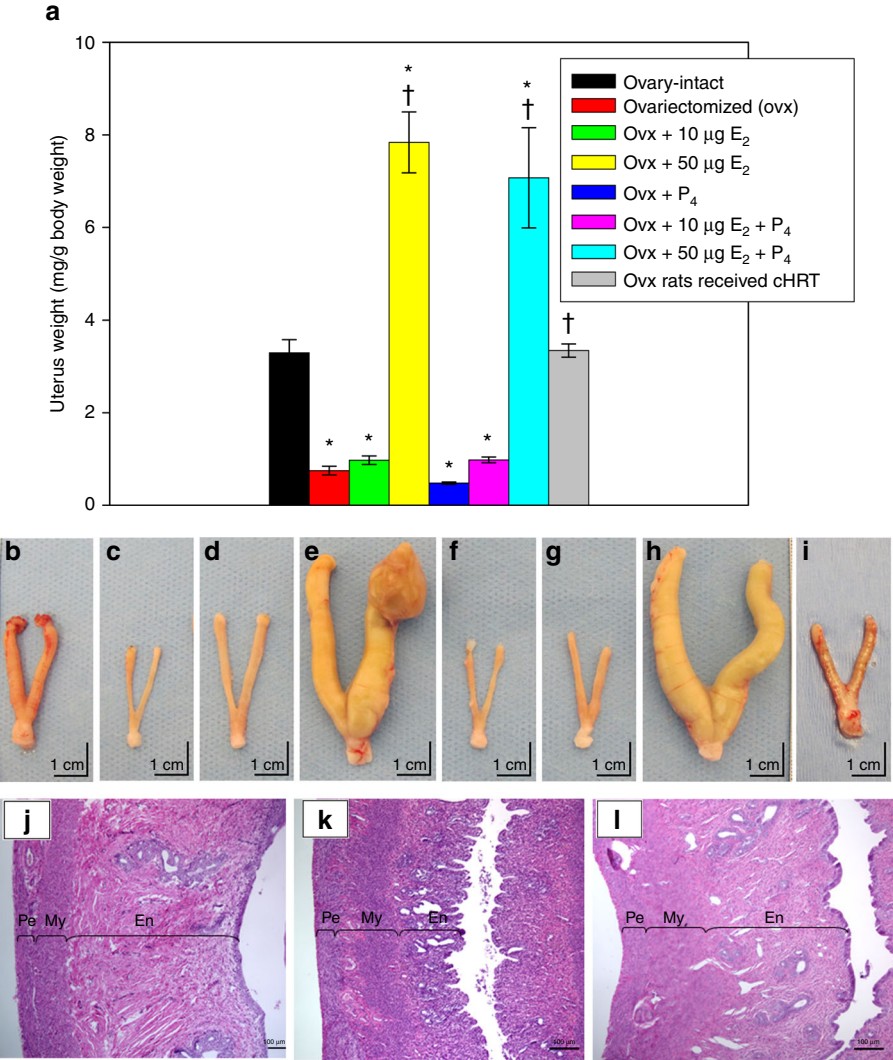

**Fig. 6** Uterine morphometry in sham and ovx rats with or without treatment. **a** Uterine weight in the ovx rats that received hormone pellets or the bioengineered constructs. **b–i** (scale bars represent 1 cm) show the gross explants, which reflect the data from **a**. **b** ovary-intact; **c** ovx; **d** ovx + 10 μg $E_2$; **e** ovx + 50 μg $E_2$; **f** ovx + $P_4$ only; **g** ovx + 10 μg $E_2$ + $P_4$; **h** ovx + 50 μg $E_2$ + $P_4$; **i** ovx rats that received cHRT. Each bar represents the mean ± SEM of 10 animals (biological replicates) with each data point as a single technical experiment. * indicates significance $p < 0.05$ compared to ovary-intact rats; † indicates significance $p < 0.05$ compared to ovx rats as determined by one-way ANOVA followed by Student–Newman–Keuls post hoc analysis. Histology of uterus demonstrating the preservation of the endometrial layer of uterus after the implantation of ovarian construct (**l**) compared to ovary-intact rats (**j**) and ovx control (**k**). En, endometrium; My, myometrium; Pe, perimetrium. **j–l** scale bars represent 100 μm

uterine morphology. Following euthanasia of animals at 90 days after the treatment period (in ovx animals treated with pHRT or cHRT), uterine tissue was excised from all groups of rats, weighed and used for histological analysis. As shown in Fig. 6a, e, h, the effects of the high-dose estrogen (50 μg $E_2$ + $P_4$) were profound, leading to extreme increases in uterine weight (Fig. 6a) attributable to the hypertrophy shown in the gross images (Fig. 6e, h). The low-dose estrogen pHRT (10 μg $E_2$ + $P_4$) formulations (Fig. 6a, d, g) had virtually no effect on uterine weight, compared to untreated ovx animals (6a-red color bar and 6d). In contrast, implantation of the bioengineered constructs led to uterine weight demonstrably higher than the ovx animals despite plasma hormone levels essentially similar to the low-dose $E_2$ pHRT (Fig. 6a gray color bar and 6i). Indeed, the uterine weight achieved with the construct was not different from that of the ovary-intact group. Also, in contrast to the high $E_2$ pHRT doses (50 μg $E_2$ ± $P_4$), no uterine hypertrophy was observed in the bioengineered construct-treated animals (compare Fig. 6e or 6h to 6i for high-dose pHRT and the cHRT, respectively). In fact,

as shown in Fig. 6e, evidence of hyperplasia and possibly tumor growth was observed with the high-dose estrogen regimens (50 μg $E_2$ ± $P_4$), consistent with concerns noted in human patients on estrogen therapy[27, 28]. These results from the high $E_2$ pHRT dose treatment groups are also consistent with previous reports from animal studies showing extensive hypertrophied tissue formation at high $E_2$ doses[29, 30] and highlight the challenges of balancing safety and efficacy in pHRT treatments, albeit, our data generated in a rat model remains to be validated in human studies. In addition, since we did not assess the effect of an intermediate estrogen dose regimen, it remains to be determined if such a dose would eliminate the adverse effects of the high estrogen dose observed in our studies in the ovx rat model.

Histological evaluations of the uterine tissue revealed that bioengineered constructs (Fig. 6l) restored or maintained the endometrial layer of uterus (noted as En in Fig. 6j through l) that was atrophied in ovx rats (Fig. 6k) to an appearance essentially similar to uterine tissue of ovary-intact rats (Fig. 6j). Importantly, the endometrium of the ovx rats that received the ovarian

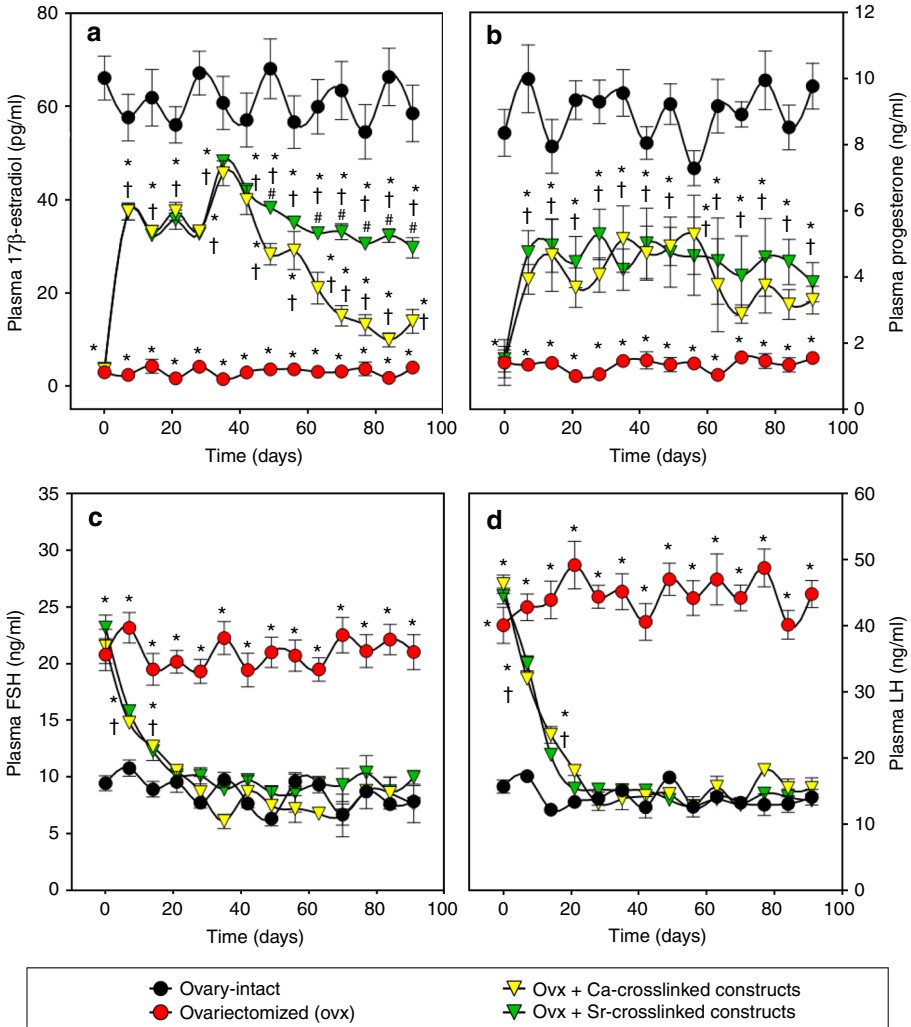

**Fig. 7** Plasma hormone levels in sham and ovx rats treated with constructs cross-linked with $Sr^{++}$ or $Ca^{++}$. Levels of $E_2$ (**a**), $P_4$ (**b**), FSH (**c**) and LH (**d**) in the ovx rats that received cHRT constructs cross-linked with either $Ca^{++}$ or $Sr^{++}$. Following a decline in the levels of $E_2$ observed in Fig. 2c, it was suspected that calcification of the constructs as evident from Fig. 1d might be the cause. Therefore, the study (Figs. 2 and 3) was repeated by replacing $Ca^{++}$ with $Sr^{++}$ and endocrine functions of modified constructs were assessed. The replacement of $Ca^{++}$ with $Sr^{++}$ improved the $E_2$ secretion by the construct as is evident from the sustained stable hormone release by the $Sr^{++}$-cross-linked cHRT constructs (green) compared to $Ca^{++}$-cross-linked cHRT constructs (yellow; see also the green data points in Fig. 2c for cHRT constructs with $Ca^{++}$-crosslinking). Each point represents the mean ± SEM of 10 animals (biological replicates) with each data point as a single technical experiment. * indicates significance $p < 0.05$ compared to ovary-intact rats; † indicates significance $p < 0.05$ compared to ovx rats; # indicates significance $p < 0.05$ compared to $Ca^{++}$-cross-linked construct as determined by one-way ANOVA followed by Student–Newman–Keuls post hoc analysis

constructs showed no signs of hyperplasia or hypertrophy. Taken together, these results indicate that the levels of sex steroid hormones produced by the bioengineered constructs were physiologically relevant. The implantation of the bioengineered constructs resulted in morphology comparable to normal animals, even with their delivery of lower plasma steroid hormone levels, and without any adverse effects as was observed with the high-dose $E_2$ pHRT treatment.

**Approach to achieve sustained secretion of estrogen**. At the conclusion of the studies reported in Figs. 2–6, we observed that plasma levels of $E_2$ decreased over time beginning at about day 40 through the remainder of the 90 days period of study (see Fig. 2c). Based on the results shown in Fig. 1d, we considered that one possible reason for this decrease was calcification of the constructs due to the use of calcium as the alginate crosslinker. To rectify this

problem, we conducted a follow-on study in which the alginate in the cHRT constructs was cross-linked with strontium ($Sr^{++}$) rather than calcium ($Ca^{++}$). In this second set of experiments, we again used ovx rats as the negative control and ovary-intact animals as a positive control to maintain an internal reference to the previous studies. As shown in Fig. 7a, and consistent with Fig. 2c, estrogen levels declined over time when $Ca^{++}$ was used as the ionic crosslinker. However, the use of the $Sr^{++}$ crosslinker abolished the decline in estrogen levels that occurred between 60 and 90 days after the $Ca^{++}$ cross-linked construct implantations. The results for progesterone, FSH, and LH in this follow-on study (Fig. 7b–d) were consistent with those shown in Fig. 2d (progesterone), Fig. 3c (FSH), and Fig. 3d (LH) with constant hormone production over the course of time and no apparent change in the levels of these hormones when strontium was used in place of calcium. Thus, the replacement of $Ca^{++}$ with $Sr^{++}$ eliminated construct calcification, as shown in the explant (Supplementary Fig. 2) and allowed

maintenance of the secreted levels of plasma estradiol (Fig. 7a). We also note that the sustained levels of estrogen secretion likely indicate minimal luteinization of the granulosa cells while sustained LH levels indicate active theca cell function.

## Discussion

Tissue or cell encapsulation may offer effective strategies to fabricate ovarian constructs for the purpose of fertility and/or hormone replacement. Approaches using segmental ovarian tissue or whole-follicle implantation (typically with a focus on cryopreservation of the tissue for reproductive purposes) have resulted in detectable hormone levels in the blood after transplantation[31]. Previous studies have also shown that autotransplantation of frozen-thawed ovarian tissue can lead to hormone secretion for over 5 years in humans[32], albeit, there is a possibility that those individuals may have had some residual hormone secretion. Also, whole-follicle encapsulation leads to the growth of antral follicles and secretion of pre-ovulatory levels of estradiol[33–35]. It is well established that high levels of estradiol secreted by pre-ovulatory follicles induce LH surge via a positive feedback[36], which in turn would induce ovulation of the encapsulated follicles with concomitant luteinization of the follicular cells. Given the added challenges of whole-follicle encapsulation for large mammalian species[37], cell-based approaches are appealing for hormone replacement because of the ease of cell encapsulation. In addition, a recent study has demonstrated that a bioprosthetic ovary created using 3D printed microporous scaffolds restored ovarian function in sterilized mice[38]. Although these approaches can be used to achieve the dual purpose of fertility and hormone replacement in premenopausal women undergoing premature ovarian failure, they would have limited application in postmenopausal women who only need hormone replacement to manage menopausal symptoms and in whom fertility is not desirable. In full development, the technology we describe in this paper, which is focused on hormone replacement, would meet the needs of the latter group of women.

Cell-based approaches have included encapsulated mixed cell cultures[39] or used co-culture approaches to ovarian cell encapsulation for cHRT[15, 40, 41]. Sustained hormone secretion in vitro and in vivo[40, 41] and amelioration of osteoporotic bone architecture in vivo[39] have been reported. We have previously reported the development of a 3-D cell encapsulation approach to mimic native structure-function relationships through multilayer bioengineered ovarian cell constructs as a means to deliver cell-based HRT (cHRT)[15]. These multilayer constructs were shown to have enhanced hormone secretion in vitro compared to co-cultures, but the in vivo behavior of the constructs remained to be determined. As a first translational step for this cHRT design, we have investigated plasma levels of several ovarian hormones, pituitary gonadotropins, and physiological end-organ effects of the cell-based constructs.

For all studies in this report, isogeneic cells obtained from donor animals were used to fabricate constructs that were implanted in the ovariectomized rats of the same strain as the donors. The construct presents differential compartmentalization of the two endocrine cells, mimicking the native follicle. The design shows two alginate layers (each containing a different cell type) separated by a poly-L-ornithine (PLO) layer which could provide immuno-isolation for the granulosa cells, but the practical purpose of this PLO coating was to provide a basement membrane for the outer layer of alginate where the theca cells were encapsulated. An additional outer coating with PLO on top of the theca cell layer, which provides immuno-isolation for both the granulosa and theca cells in the construct was added prior to a final coating of the construct with alginate. Although we

investigated only isogeneic cell-based constructs, this design could be used in the future to test the suitability of our cHRT constructs with either allogeneic or xenogeneic cells since we and others have previously shown that a PLO-coated alginate microcapsule system provides immuno-isolation of both allogeneic and xenogeneic cells after transplantation[42, 43].

In our studies, we compared the performance of the cHRT constructs with an experimental pHRT treatment regimen in the ovx rat model. There are numerous clinical delivery methods for pHRT including oral, transdermal patches or gels, injections, creams, and subcutaneous implants. While the pharmacokinetics of these methods vary[44, 45], the goal with these clinical approaches is to achieve a steady-state concentration of the hormones in the plasma in a range effective for an individual patient. Using subcutaneous pHRT delivery we were able to achieve two different levels of circulating estrogen levels, which we have denoted as low- and high-dose estrogen regimens. While the low-dose estrogen is physiologically relevant, the high-dose is supraphysiological. Importantly, the low-dose regimen resulted in plasma concentrations of estrogen and progesterone similar to those achieved in other reports[46, 47] and by our cHRT constructs, allowing direct comparison of pHRT and cHRT. While the high-dose regimen we used would not be clinically acceptable due to safety issues, this dose has been used in previous studies to achieve beneficial effects[20, 22], and in our current studies has demonstrated that supraphysiological plasma levels of estrogen were required for pHRT to achieve benefits in bone health that were comparable to those achieved by cHRT at much lower plasma hormone concentrations.

The systemic effects achieved by the cell constructs were made possible through their sustained sex steroid hormone production along with their placement in the vascularized omentum. The omentum is considered a clinically-translatable implant site for implantation of tissue engineered constructs because of its vascular supply to the implants[48], as apparent in the engineered tissue explants in our studies, and particularly, has been shown to provide a better site, in terms of follicle survival, for intra-abdominal ovarian cortex transplantation in the baboon compared to other sites[42]. Plasma levels of $E_2$ and $P_4$ in animals treated with our bioengineered constructs were not as high as those measured in the ovary-intact rats. However, hormone levels in the ovary-intact animals are of reproductive significance whereas the goal of cHRT (or pHRT) is to achieve other physiological benefits at the lowest plasma levels possible. The LH levels measured after low-dose estrogen with progesterone pHRT and cHRT construct implantation were equivalent to those of ovary-intact animals, consistent with previous reports indicating that $E_2$ at low doses with $P_4$ suppressed LH levels[49]. We note that the FSH levels in ovx (no treatment) were approximately 2.5-fold higher than ovary-intact in our studies, which used ELISA for hormone measurement. Others have reported much higher FSH levels after ovariectomy compared to ovary-intact rats in studies using the more sensitive radioimmunoassay for hormone measurement[50]. Thus, the disparity in the detection limits likely explains the differences in the magnitudes of LH elevation following ovariectomy. Since ovarian sex steroid hormones are known to be synthesized and secreted under the stimulation of FSH and LH[13, 14], the elevated levels of these gonadotrophins after ovariectomy in the recipient rats could have acted as the trigger for the secretion of sex steroid hormones from the implanted constructs. It is also possible that even small numbers of oocytes may play a role in granulosa cell secretion of estrogen[51], but we cannot rule out the possibility that hormone secretion may also occur in the absence of oocytes[52].

The plasma levels of estradiol and progesterone achieved by the cHRT constructs, though lower than the levels in ovary-intact

rats, were effective in suppressing the levels of FSH and LH in ovx rats. The increased levels of inhibin in the plasma of ovx rats that received the construct implants validates our previous in vitro findings of the secretion of activin and inhibin by the cHRT constructs[15]. Given the known effects of estrogen and inhibin on the hypothalamus secretion of gonadotrophin releasing hormone (GnRH), our interpretation of these results is that the cHRT constructs re-established the HPO axis in the ovx animals, in contrast to the pHRT. It is presently unclear if other molecules such as kissipeptin, which acts upstream of GnRH in the sex steroid feedback mechanisms[53], participate in the regulation of the HPO loop mediated by our constructs. These observations reinforce the notion that the concentrations of ovarian sex steroid hormones secreted by the ovarian constructs were physiologically relevant and achieved efficacious end-organ results at lower and safer plasma levels. The secretion of other ovarian hormones (e.g., inhibin) as well as androgen precursors (e.g., testosterone) of female steroid hormones by our constructs, which are not present in pHRT regimens, has wider implications. For example, it has been reported that osteoporosis associated with ovarian failure is not solely due to the decline in sex steroids, but may also involve decline in inhibins as well as elevated levels of FSH. Elevated FSH following disruption of the negative feedback loop is known to induce bone loss by increasing osteoclastogenesis[54]. The secretion of hormones such as inhibin (Fig. 3e) may also explain our results showing maintenance of low levels of FSH (Fig. 3c) despite decreasing secretion of estrogen (Fig. 2c). These examples highlight the significance of other hormones and factors secreted by ovarian endocrine components, which can only be achieved through cHRT.

The results of our follow-on study clearly show that the replacement of $Ca^{++}$ with $Sr^{++}$ in the alginate crosslinking corrects the problem of time-dependent decline in estrogen secretion from the constructs seen in our first set of experiments where $Ca^{++}$ was used for alginate crosslinking. We suggest a couple of possibilities for the abolition of the time-dependent decline in estrogen levels from constructs fabricated using alginate crosslinked with $Sr^{++}$. First, when $Ca^{++}$ is used to crosslink alginate it precipitates out of the hydrogel matrix over time with consequent calcification of the constructs. Alternatively, it is possible that the alginate matrix in the construct in which $Ca^{++}$ is used for crosslinking may provide a microenvironment with higher stiffness[55] than is appropriate for sustained function of the encapsulated ovarian cells, thus causing a decline in construct performance over time. Although not tested directly, the results with the $Sr^{++}$ crosslinking in which estrogen levels were maintained at steady levels indicates that luteinization of the granulosa cells did not occur[56].

In summary the cell-based system of hormone replacement described in this report offers an attractive alternative to traditional pharmacological approaches and is consistent with current guidelines in the U.S. and Europe recommending the lowest possible doses of hormone for replacement therapy[57, 58]. In our studies plasma levels of estrogen and progesterone (from our cHRT constructs) similar to the recommended levels led to improved end-organ outcomes, thus demonstrating the ability of cHRT constructs to achieve the desired benefits. Sustained stable hormone release over the course of 90 days of study was demonstrated by our constructs when using the $Sr^{++}$ crosslinker. We also demonstrated effective end-organ outcomes in body fat composition, uterine health, and bone health. However, additional studies will be required to determine the sustainability of the hormone secretion of the $Sr^{++}$-cross-linked constructs by measuring hormone levels from implanted constructs for periods longer than 3 months such as 6–12 months in the rat model.

In conclusion, this study highlights the potential utility of cHRT for the treatment and study of conditions associated with functional loss of the ovaries. Although longer-term studies would be of future interest, the 90-day duration of this rodent model study is consistent with others investigating osteoporosis in an ovariectomy model[59–62]. However, while our study provides a proof-of-concept for cHRT, it suffers the limitation that it is only an isogeneic-based construct implantation. We concede that further studies in either allogeneic or xenogeneic settings would be required with the construct design described in this report in the path towards clinical translation given that patients who would receive this type of treatment are unlikely to have sufficient autologous ovarian cells for transplantation.

## Methods

**Animals**. All animal studies were conducted with the approval of the Wake Forest University Health Sciences Animal Care and Use Committee. Female Fischer 344 rats purchased from Harlan Sprague-Dawley Inc. (Indianapolis, IN) were used in this study. 21-day old immature female rats were used as donors for the isolation of granulosa and theca cells, while 3-month old mature female rats served as bioengineered ovarian cell construct recipients. These studies were conducted with 3 cohorts of animals with a total of 14 treatment groups. All 3 cohorts had sham-operated ovary-intact rats and ovariectomized (ovx) rats as internal cohort controls. The first cohort consisted of seven groups, including the following (1) sham-operated ovary-intact rats, (2) ovx rats, (3) progesterone only (2 mg/kg), (4) low-dose estrogen (10 μg/kg), (e5) low-dose estrogen (10 μg/kg) with progesterone (2 mg/kg), (6) high-dose estrogen (50 μg/kg) estrogen or (7) high-dose estrogen (50 μg/kg) with progesterone (2 mg/kg). All animals receiving pHRT treatments in the first cohort (listed 3–7above) received the hormones via a subcutaneous implanted pellet as described below and the ovx animals received placebo subcutaneous pellets instead of pellets loaded with hormone. The second cohort consisted of 3 groups, including (1) sham-operated ovary-intact rats, (2) ovx rats that received blank capsules without any cells, (3) ovx rats implanted with constructs in which alginate was cross-linked with $Ca^{++}$. The third cohort consisted of four groups including: (1) sham-operated ovary-intact rats; (2) ovx rats that received blank capsules without any cells; (3) ovx rats implanted with constructs in which alginate was cross-linked with $Ca^{++}$; or (4) ovx rats implanted with constructs using $Sr^{++}$-cross-linked alginate. Animals that exhibited any unrelated pathological conditions that would have affected the study outcome were excluded from the study based on pre-established conditions included in the original animal protocol.

**Materials**. Medium 199 and McCoy's 5 A media were purchased from Gibco-BRL (Life Technologies/Gibco-BRL, Grand Island, NY). Percoll, FSH, LH, insulin-transferrin-selenium mix (ITS), deoxyribonuclease I (DNase 1), poly-L-ornithine (PLO molecular weight 15–30 KDa) were purchased from Sigma-Aldrich (St. Louis, MO). Low viscosity (20–200 mPa·s) ultra-pure sodium alginate with high mannuronic acid (LVM) content and with high guluronic acid (LVG) were purchased from Nova-Matrix (Sandvika, Norway). Collagenase type 1 was from Worthington (Lakewood, NJ) and insulin-like growth factor-I (IGF-I) from Peprotech (Rocky Hill, NJ). Solutions for alginate microcapsule synthesis were made using the following chemicals: HEPES, sodium chloride, calcium chloride, and strontium chloride (Fisher Scientific, Pittsburgh, PA). The vendors for other chemicals, reagents, assay kits and antibodies used have been indicated in the relevant areas of this method section.

**Cells isolation and purification**. The endocrine cells were isolated from ovaries of immature rats[15]. The ovaries collected in ice-cold medium 199 (M199) [containing HEPES (25 mM), 1 mg/ml bovine serum albumin (BSA), L-glutamine (2 mM), penicillin (10, 000 IU/ml), streptomycin (10, 000 μg/ml), and amphotericin B (25 μg/ml)] were cleansed of extraneous tissues, washed twice with ice-cold M199 and punctured gently with 27 G syringe needles in order to release the loosely packed granulosa from the follicles; cells thus collected were kept on ice. The remaining ovaries were chopped into fine pieces of ~ 0.25 mm² and the cells released during this process were also collected and kept separately on ice. The pieces of ovaries were then incubated with collagenase (2 mg/ml) and DNase (10 μg/ml) in M199 for 90 min with occasional mixing. The enzyme-digested pieces were dispersed by using a Pasteur pipette to obtain a single cell suspension and stored on ice as a separate fraction. Using different fractions collected from the above mentioned steps granulosa and theca cells were purified by a discontinuous Percoll gradient separation and the purity of each cell type was assessed by flow cytometric analysis using cell-specific markers (CYP19 for granulosa cells and CYP17A1 for theca cells). The viability of the cells assessed using the Trypan blue method was found to be between 85–95%, indicating only little batch-to-batch variability.

**Culture of granulosa and theca cells**. Purified granulosa and theca cells were cultured in McCoy's 5 A medium [with 2 mM L-glutamine, 1 mg/ml BSA, 10,000 IU/ml penicillin, 10,000 μg/ml streptomycin, 25 μg/ml amphotericin B, ITS (10 μg/ml insulin; 5.5 μg/ml transferrin; 5 ng/ml selenium; 0.5 mg/ml BSA; 4.7 μg/ml linoleic acid) and 10 nM IGF-I] with supplementation of different hormones to maintain their respective phenotypes. While the media for granulosa cell culture were supplemented with 50 ng/ml FSH and 100 nM $E_2$ the media for theca cells were supplemented with 50 ng/ml LH and cultured at 37 °C under an atmosphere of 5% $CO_2$ in humidified air until they reached 80%–90% confluency. Upon reaching 80%–90% confluent stage (~ 96 h), the granulosa and theca cells were lifted from the culture flask with a cell scrapper and used for studies.

**Ovarian tissue-construct encapsulation**. The ovarian tissue-constructs that mimic the follicular architecture were fabricated by using encapsulation techniques[15]. Briefly, for fabricating the cHRT construct, first the granulosa cells were encapsulated in 1.5% (w/v) ultra-pure low viscosity mannuronic acid (up-LVM) and coated with 0.1% (w/v) poly-L-ornithine (PLO) for 20 min. The PLO-coated microcapsules were then mixed with theca cells suspended in 1.5% (w/v) up-LVM and encapsulated again using the micro-fluidic device in order to obtain double alginate-layered microcapsules. An additional layer of PLO followed by a layer of 0.5% (w/v) ultra-pure low viscosity guluronic acid (up-LVG) purchased from Nova-Matrix (Sandvika, Norway) were then applied on the periphery. The additional layers of PLO and alginate were included as this design would be appropriate for immune-isolation of the encapsulated theca cells in the external compartment of the construct (Fig. 1b) should allogeneic or xenogeneic cells be used. The cHRT construct design has structural architecture resembling that of native follicles as illustrated in Fig. 1a. The differential distribution of ovarian endocrine cells in the multi-layered microcapsules was demonstrated by encapsulating pre-staining granulosa and theca cells with Cell Tracker green and Cell-tracker Orange (Life Technologies/Invitrogen, Grand Island, NY), respectively. The multilayer microcapsules were then imaged by confocal microscopy (Zeiss LSM510) to visualize the compartmentalization of each cell type, as depicted in Fig. 1c.

**Bilateral ovariectomy transplantation of bioengineered ovarian cell constructs or delivery of pharmacological HRT**. Female Fisher *344* rats aged 3 months (weighing 208.2 ± 19.7 g) were subjected to bilateral ovariectomy (ovx). Blood samples were collected weekly (between 8 and 11 am) in order to follow the changes in plasma levels of $E_2$, $P_4$, FSH and LH in these rats (see below for methods of measurements). After confirming that the circulating $E_2$ and $P_4$ had reached their baseline, the ovx rats were divided into three cohorts (7 groups in the first cohort, 3 groups in the second cohort and 4 groups in the third cohort) comprising a total of 14 groups ($n = 10$/group) where the animals were randomly assigned within each cohort. Animals typically were allowed to reach stable baseline plasma levels of $E_2$, $P_4$, at 4–6 weeks post ovariectomy, and treatment was begun thereafter. The number of animals was derived from a Power calculation based on our preliminary data for differences in means, standard deviations, and assuming power = 0.8 and alpha = 0.05.

Following analysis of the plasma levels of $E_2$, $P_4$, FSH, and LH, we selected five different pHRT treatments in an effort to mimic or exceed the plasma levels achieved with the bioengineered ovarian cell constructs and to simulate the temporal plasma levels that might be obtained in pHRT delivery. These five pHRT groups were: (1) progesterone only (2 mg/kg), (2) low dose of estradiol only (10 μg/kg), (3) high dose of estradiol only (50 μg/kg), (4) low dose of estradiol + progesterone (10 μg/kg and 2 mg/kg, respectively), and (5) high dose of estradiol + progesterone (50 μg/kg and 2 mg/kg respectively). The sixth and seventh groups in the first cohort were the sham-operated (ovary-intact) and ovx (with placebo pellets not containing hormones) controls, both of which acted as internal controls for comparison to the second cohort (below). For pHRT the given estrogen and/or progesterone doses were delivered by pellets (Innovative Research of America, Sarasota, FL) implanted subcutaneously as described by others previously[17, 20, 63–66]. The pellets were for 90-day release to match the time scale of the measurements taken for the cHRT constructs and contained 17β-estradiol, progesterone, or nothing (placebo control).

In the second cohort, one group of ovx rats had constructs implanted in their omentum as a pouch. A second group of ovx rats received blank microcapsules without any cells (untreated ovx group), and the third group was sham-operated (ovary-intact). In the third cohort, two groups of ovx rats were implanted in their omentum pouches with either $Ca^{++}$-cross-linked or $Sr^{++}$-cross-linked cHRT constructs. A third group of ovx rats that received blank microcapsules without any cells (untreated ovx group) and the fourth group consisting of sham-operated, ovary-intact rats served as controls for the third cohort. The construct transplant recipient groups were implanted with tissue-constructs containing ~ $5 \times 10^6$ cells of each type in the omentum pouch. We estimate that there are ~ 2500 of each cell type (granulosa and theca) for a total of 5000 cells in each construct. We implanted approximately 1000 constructs per animal for a total of $5 \times 10^6$ cells. The number of cells per construct is much lower than in an individual follicle (~ 200,000 granulosa cells)[67], but the total number of implanted cells is similar to the number of cells in each ovary of the rat ($4.7 \times 10^6$ of each cell type)[68]. The body weights of the rats were measured during weekly blood sample collections (between 8 and 11

am to measure plasma levels of hormones ($E_2$, $P_4$, testosterone, FSH, LH and inhibin) for 90 days.

**Measurement of plasma 17β-estradiol progesterone testosterone FSH LH and inhibin**. To evaluate the endocrine functions of implanted tissue-constructs in the ovariectomized rats, the levels of $E_2$, $P_4$, FSH and LH were measured in the plasma samples obtained from all fifteen experimental groups of rats. Levels of testosterone and inhibin were measured only in the plasma samples obtained from the cohort 2 rats. The levels of sex steroid hormones in the plasma samples were measured using ELISA kits following the manufacturers' instructions. The $E_2$ was measured by competitive ELISA kits purchased from Enzo Life Sciences (Plymouth Meeting, PA; ADI-901-174) and is reported to have 17.8% cross reactivity with estrone ($E_1$), 0.9% cross reactivity with estriol ($E_3$), and less than 0.5% cross reactivity with other steroid hormones compared to 100% for $E_2$. The $P_4$ was also measured by competitive ELISA from Enzo Life Sciences (ADI-901-011) and is not reported to have cross reactivity with other steroid hormones. The testosterone was measured by competitive ELISA kits purchased from Enzo Life Sciences (Plymouth Meeting, PA; ADI-901-065) and is reported to have 14.6% cross reactivity with 19-hydro-xytestosterone, 7.20% cross reactivity with androstendione, 0.72% cross reactivity with dehydroepiandrosterone, 0.40% cross reactivity with $E_2$ and less than 0.001% cross reactivity with other steroid hormones compared to 100% for testosterone. FSH (R6403) and LH (R6316) were measured by sandwich ELISA kits from TSZ ELISA (Framingham, MA) and have no reported cross-reactivities. Inhibin was measured by competitive ELISA kits purchased from BlueGene Biotech (Shanghai, China) and has no reported cross-reactivity.

**Measurement of plasma osteocalcin and c-telopeptide of collagen**. To assess the restoration of balance in bone metabolism after cell-based HRT (cohort 2), the levels of bone turnover markers, osteocalcin (OCN) (bone formation marker) and c-telopeptide of collagen (CTx) (bone resorption marker), were measured in the plasma samples from all experimental groups of rats in the second cohort (cHRT). Both OCN and CTx in the plasma samples were measured using competitive immunoassay kits purchased from MyBioSource, Inc. (San Diego, California, USA) following the manufacturer's instructions.

**Integrity of the constructs in the explants of omentum pouches**. 90 days post-transplantation, the omentum pouches were resected out. After a gross examination for any fibrotic tissue around the omentum pouch, the cHRT constructs were retrieved from the omentum pouch and the constructs were imaged under phase-contrast microscope. A part of the omentum pouch was used to assess the viability of the encapsulated cells as described below for "Live/dead analysis of cells". Another part of the omentum pouches was fixed in 4% paraformaldehyde, washed after 24 h, processed and embedded in paraffin blocks. The blocks were sectioned at 5 μm thickness by using a Leica RM 2265 and stained with hematoxylin and eosin. The cross sections of omentum pouches were assessed for the integrity of the constructs and fibrotic capsules around the constructs at 100 x magnifications.

**Live/dead analysis of cells in the constructs after 90 days in vivo**. The viability of the encapsulated cells in the cHRT constructs was assessed using a live/dead assay[15]. Briefly, the constructs retrieved form the omentum pouches were transferred to a 24-well plate and incubated with 25 μM CFDA SE (carboxyfluorescein diacetate, succinimidyl ester) (Life Technologies/Invitrogen, Grand Island, NY) in serum-free medium for 15 min at 37 °C under an atmosphere of 5% $CO_2$ in humidified air. Then the CFDA SE containing medium was replaced with medium containing 10% FBS and incubated again under the same conditions for another 30 min. The serum-containing medium was then replaced with 50 μg/ml of propidium iodide (PI) (Life Technologies/Invitrogen, Grand Island, NY) and incubated at room temperature for 2 min and the microcapsules were washed to remove excess PI. The microcapsules were then observed under inverted fluorescence microscope and imaged. The number of live and dead cells was determined qualitatively from the composite image acquired using Image-Pro plus software (version 6.3.1.542).

**Dual-energy X-ray Absorptiometry and micro-computed-tomography scans**. Dual-energy X-ray Absorptiometry (DXA) measurement was performed by using a Hologic QDR-4500 Elite Fan Beam X-ray densitometer (Hologic Inc., Waltham, MA, USA). Fat distribution and whole body bone mineral density (BMD) in vivo were measured in anesthetized rats using a small animal software (version 13.3:3) according to methods described by Rosen et al.[69]. Femur bones were collected and their qualities were analyzed at Biomedical Research Imaging Center (University of North Carolina, Chapel Hill, NC) by using cone-beam X-ray micro-computed-tomography (μCT) (Scanco Medical AG, Bruttisellen, Switzerland). The bones were immersed in saline solution (0.9% NaCl) and the distal region of the bone was scanned at a slice thickness of 18 μm. Three-dimensional images were reconstructed using the SCANCO-40 software and the following parameters were analyzed: BMD, fractional trabecular bone volume (BV/TV [%]), trabecular thickness (Tb.Th [μm]), Trabecular number (Tb.N [1/mm]), trabecular separation (Tb.Sp [μm]).

**Uterine weight and histology**. Uterine tissue was explanted from each rat at the end of the study, imaged at the gross level, weighed and fixed in 4% paraformaldehyde. After 24 h the tissues were washed, processed and embedded in paraffin blocks. The blocks were sectioned at 5 μm thickness by using a Leica RM 2265 and stained with hematoxylin and eosin. The longitudinal sections of uteri were assessed for the thickness of (a) perimetrium [Pe]; (b) myometrium [My] and; (c) endometrium [En] at 100 x magnifications.

**Statistical analysis**. The investigator conducting measurements used for statistical analysis was blinded to the experimental groups for each study. Results are presented as the mean ± standard error of the mean (SEM) unless stated otherwise. Comparisons between different groups were performed by using one-way analysis of variance (ANOVA) followed by Student–Newman–Keuls post hoc analysis. Differences were considered to be statistically significant at $p < 0.05$. Statistical analyses were performed using SPSS software (version 10.0.1). Satisfaction of model assumptions and best estimate of error was determined in SPSS software.

**Data availability**. The authors declare that all data supporting the findings of this study are available within the article and its Supplementary Information files or from the corresponding author upon reasonable request.

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

## Acknowledgements

This study was supported, in part, by Jack and Pamela Egan. The study was also supported in part by the National Institutes of Health grants R01DK080897 (ECO), R01AR061391 (J.M.S.), T32EB014836 (J.P.M.; PI: A.A.), but the content of this article is solely the responsibility of the authors and does not necessarily represent the official views of the National Institutes of Health. We also would like to thank, Juan Arenas, Nathaniel T. Marshall, Myung Jae Jeon, Pareta R, Cathy M. Mathis, Cynthia Zimmerman, and Victoria A. Lipitt for their technical assistance in the study.

## Author contributions

S.S.: Participated in designing the study, conducted the experiments, analyzed the data, and wrote the manuscript. J.M.S.: Participated in designing the study, obtained Institutional Animal Care and Use Committee approval, interpreted the data, and wrote the manuscript along with S.S. J.P.M. and S.J. assisted with the animal surgeries and the data collection. T.C.R.: Performed the DXA scans, data analysis and provided expert opinion. J.J.Y. and A.A.: Conceived the idea, provided their expertise and feedback. E.C.O: Conceived and designed the study, supervised the experiments, edited and revised manuscript. All authors read and contributed critically to the manuscript and gave final approval for publication.

## Additional information

**Competing interests:** The authors declare that there are two patents associated with the technology described in this paper. The patents are (1) E.C.O., J.J.Y., J.M.S., S.S., A.A. US Patent # 9,283,251 B2; Encapsulated cells for hormone replacement therapy (issue date March 15, 2016) and (2) E.C.O., J.J.Y., J.M.S., S.S., A.A. US Patent # US 9,763,986 B2. Encapsulated cells for hormone replacement therapy (issue date 9/19/2017). The remaining authors declare no competing financial interests.

