## [Peer Review File · Nature Communications]

Referee: Development of an improved bio-mimetic ovarian cell construct for controlled hormone delivery and correction of abnormalities associated with ovarian failure

Very interesting proof of concept but the main concerns are quite numerous and a paragraph on limitations should be added. Agree for editorial.

Comments

The manuscript described the “in-vivo” performance of 3D bioengineered ovarian constructs: 2 types of constructs were used AICC and AXCC.

The authors reported secretion of hormones by the constructs and their subsequent effects on pituitary, bone, uterus and fat tissue in comparison with p(pharmacological) HRT.

Although the proof of concept is very interesting, I have some main concerns:

- 1) The abstract mentioned that the constructs achieved long-term secretion, which is not the case; at least for the Ca-crosslinked constructs.
- 2) The proof of concept is correct for isogenic cells, but there are no data concerning the efficacy of AXCC containing allogenic GC and TC cells.
- 3) There is no proof (last sentence) that the constructs are more effective than pHRT. Moreover, guidelines concerning pHRT recommend the lowest effective dose.
- 4) Fig 1 shows E₂ secretion, no longer than 50 days (Ca-crosslinked constructs). How explain that FSH remains low while E₂ levels are similar to ovariectomized rates (70-80 days).
- 5) Some sentences (lines 244-269) concerning pHRT and endometrial safety are not correct.
- 6) Increased incidence of breast cancer in women taking pHRT is due to progestogens. What is the rationale to add TC? Why not a construct containing only GC?
- 7) Concerning the bone, 90 days is a very short period.
- 8) This referee would like to have data from Sr-crosslinked constructs for a long period. Secretion from constructs for a period of 3 months is too short. Please discuss and maybe make reference to the paper of Donnez and Dolmans (JARG), as these authors demonstrated that the duration of ovarian secretion is as long as 5 years after reimplantation of cryopreserved ovarian tissue.
- 9) The list of references is not appropriate for the reasons that: 1) there are many “old” references. Concerning HRT, the recent papers of Lobo are not cited. 2) no reference on ovarian tissue reimplantation, 3) no reference on the artificial ovary.

Minor concerns:

- 1) Introduction: much too long (line 76-87) could be summarized in one sentence.
- 2) Line 104-111 are in fact a summary of the results. Results should not appear in the introduction but kept for later. At the end of the introduction this referee would like to read the aim of the study.
- 3) Line 110: safer??
- 4) Line 146-150: What is the rationale to give 50µg E₂ which led to supraphysiological levels? Low doses seem to be effective.
- 5) Data concerning bone density and parity, although reassuring, cannot be extrapolated to human.
- 6) Discussion: much too long. Too many redundant.
- 7) Methods: sentences line 528: why 344 rats if n=10/group, and 15 groups?

8) L536: Why progesterone only? (controls are already included in the study).

Reviewer #2:

Remarks to the Author:

The manuscript by Sittadjody et al. describes an in vivo strategy for the construction of a 3-dimensional ovary with granulosa cells and thecal cells for delivery of ovarian hormones. The investigators use rat as the model and place the bioengineered ovary (cell encapsulated ovarian cells) into the omentum. Circulatory steroid hormone levels were maintained at a level intermediate between intact and castrate females (Figure 2) and secondary beneficial effects of normalized hormone levels were observed (Figure 4, 5).

Overall, this is a nice story that advances the field and may have important long-term consequences on the health of aging women.

Minor point:

1. "Thecal" is an adjective; "theca" is a noun. The authors should correct to be "thecal cells" throughout.

Reviewer #3:

Remarks to the Author:

Reviewer:

The authors report a new strategy to overcome the problems of pharmacological hormone replacement therapies (pHRT) where high hormones blood levels in short periods of time may lead to adverse effects. This new strategy consists in a cell-based hormone therapy (CHTR) where hormones producing cells are embedded within a biomaterial forming capsules and subsequently, these capsules are implanted in ovariectomized rats. This approach, allow a strict hormone secretion control because the therapeutic cells are stimulated by physiological signals promoting the necessary hormone secretion minute to minute. The manuscript is -in my opinion- original, well written and complete, but some aspects are to be clarified:

1) The mechanical properties of ionically crosslinked alginate capsules can be modified (even coming to break down) due to the diffusion of Ca^{2+} ions.

Did the authors studied the stability and degree of degradation of the constructs after the experimentation process?

2) Did the authors test possible effects of the use of calcium on the release of hormones by the encapsulated cells?

Minor comments:

1) In figure 1G, I,J,K,L,M the scale bar should be bigger.

Response to Reviewer Concerns:

We thank the reviewers and the editor for their thorough critiques of our manuscript entitled “Development of an improved bio-mimetic ovarian cell construct for controlled hormone delivery and correction of abnormalities associated with ovarian failure.” We have substantially changed the manuscript to address the concerns. Below we address the points raised by each reviewer in a point-by-point manner. The reviewer comments are provided and our response is indicated in bold type. We have indicated significant changes within the document by yellow highlighting.

Reviewer #1 Comments and Critiques:

Comments:

Very interesting proof of concept but the main concerns are quite numerous and a paragraph on limitations should be added.

We have now added discussion of the limitations of our studies with these constructs in the final paragraph of the Discussion section. Other concerns are discussed below.

Major Concerns:

1) The abstract mentioned that the constructs achieved long-term secretion, which is not the case; at least for the Ca-crosslinked constructs.

We have adjusted the terminology to indicate that the release is “sustained” (see Abstract and Discussion, Paragraph 2 line 2), as “long-term” may be ambiguous. We agree that the hormone secretion with Ca-crosslinked constructs decreased with time, which highlights the significance of the follow-on study with Sr-crosslinked constructs in which no time-dependent decrease of estrogen secretion was observed.

2) The proof of concept is correct for isogenic cells, but there are no data concerning the efficacy of AXCC containing allogenic GC and TC cells.

In consultation with the editor, we have removed reference to allogeneic or xenogeneic in the description of our constructs in this revised manuscript (see revised Figure 1 and removal of AICC and especially AXCC terminology). We have very clearly noted that our manuscript provides proof-of-concept studies with isogenic cells (last paragraph of Discussion). Although we believe that the cHRT construct described in this paper is theoretically suitable for use of allogeneic or xenogeneic cells (Discussion, 3rd paragraph), we have alluded to the need for such studies in the future given that most women who would benefit from the proposed strategy would not have excess ovarian follicles that could be harvested as autografts (last paragraph of Discussion).

3) There is no proof (last sentence) that the constructs are more effective than pHRT. Moreover, guidelines concerning pHRT recommend the lowest effective dose.

We have modified the last sentence of the abstract to indicate that the constructs were more effective (body weight, uterine tissue, bone health) than the low-dose pHRT used in our study and safer (no uterine tissue hyperplasia) as was observed with the high-dose pHRT.

We fully agree that current guidelines recommend the lowest effective dose, indicated by references #81 and 82 in the revised manuscript (#77 and 78 in the original submission). Therefore, we have attempted to further clarify that a key advantage to our constructs is the ability to achieve more favorable physiological outcomes (body fat composition, uterine health, bone health) at the same plasma concentration as low-dose pHRT, which in contrast did not provide the extent of beneficial outcomes as the cHRT (penultimate paragraph of Discussion, lines 4 – 5). We have also clarified that our biomimetic constructs' ability to integrate into the hypothalamus-pituitary-ovary axis results in more favorable physiological outcomes (Discussion, Paragraph 6).

4) Fig 1 shows E2 secretion, no longer than 50 days (Ca-crosslinked constructs). How explain that FSH remains low while E2 levels are similar to ovariectomized rats (70-80 days). **The low FSH levels despite decreasing levels of E2 secretion over time in the Ca-crosslinked constructs is interesting. We have added a sentence in the Discussion (6th paragraph of Discussion, last 5 lines) and also specifically referenced Figure 3E and the role of inhibin because this data partially addresses this issue. In short, the mechanism is not fully elucidated but it is likely that this behavior is related to other factors secreted by the constructs (but not present in pHRT) including inhibin and testosterone. Unlike the E2 secretion levels from calcium constructs, the levels of inhibin and testosterone remain quite stable over the course of the 90-day study, as shown in Figures 2E and 3E. It is known that inhibin (as well as activin) and testosterone play feedback roles on the hypothalamus and pituitary, which in combination with sustained (albeit decreased) levels of E2 secretion leads to regulation of the FSH levels. In addition, other pathways such as the kisspeptin system, as indicated in our manuscript, that were beyond the scope of our study could play a role in the cHRT effects.**

5) Some sentences (lines 244-269) concerning pHRT and endometrial safety are not correct. **We have clarified the experimental groups to which we are referring in each sentence of the Results section on “Changes to Uterine Morphometry”, 2nd paragraph in order to ensure that the reader will be able to properly understand our explanation of the results. We believe that our interpretation of the literature that we have cited in regards to loss of ovarian function on the genito-urinary tract (first paragraph of “Changes in uterine morphometry” of results section) agree with our data.**

6) Increased incidence of breast cancer in women taking pHRT is due to progestogens. What is the rationale to add TC? Why not a construct containing only GC? **It is well-known that final production of estrogens from the GC require androgen production in the TC (and also that the TC require GC for progesterone used for androgen production). Highlighting this point, our previous *in vitro* work on the development of these constructs (Sittadjody et al, Biomaterials 2013) in which constructs with only GC or only TC were investigated demonstrated virtually no secretion of either E2 or P4. Thus, while progestogens can be excluded from pHRT regimens, this will not work with a cell-based (cHRT) system.**

7) Concerning the bone, 90 days is a very short period.

We believe that a 90-day time period examination of osteoporosis in rat bone following ovariectomy is highly appropriate and supported by the literature (reference #s 83-86), which we have provided in this revised manuscript (last paragraph of Discussion).

8) This referee would like to have data from Sr-crosslinked constructs for a long period. Secretion from constructs for a period of 3 months is too short. Please discuss and maybe make reference to the paper of Donnez and Dolmans (JARG), as these authors demonstrated that the duration of ovarian secretion is as long as 5 years after reimplantation of cryopreserved ovarian tissue.

As noted above for the bone study, we believe that our 90 day study period in the rat ovariectomy model is appropriate and supported by the literature. We have now cited the manuscript by Donnez and Dolmans (“Ovarian cortex transplantation: 60 reported live births brings the success and worldwide expansion of the technique towards routine clinical practice”, JARG, 2015) and have discussed key differences between their study and our own (see Discussion, first paragraph, lines 11-13).

9) The list of references is not appropriate for the reasons that: 1) there are many “old” references. Concerning HRT, the recent papers of Lobo are not cited. 2) no reference on ovarian tissue reimplantation, 3) no reference on the artificial ovary.

As noted in several of the responses above we have, where appropriate, provided newer references (though some of the “old” references are still very relevant), including those by Lobo (point 1). We have also provided additional, recent citations in the revised manuscript related to ovarian tissue reimplantation from human studies (see reference number 49), in addition to the rodent study (reference #45 in this manuscript) from the Shea and Woodruff group in the original submission (point 2). Finally, we regret that the recent manuscript by Laronda et al. was published right after (1 day) we had submitted our original manuscript thus precluding its citation in our original submission. We have now included this important reference (#50) in our revised manuscript and, in fact, we believe that our revised manuscript will provide an excellent complement to that article.

Minor concerns:

1) Introduction: much too long (line 76-87) could be summarized in one sentence.

We have reduced the length of the Introduction, as suggested by the reviewer, while maintaining key relevant information.

2) Line 104-111 are in fact a summary of the results. Results should not appear in the introduction but kept for later. At the end of the introduction this referee would like to read the aim of the study.

We have removed the summary of the results from the Introduction, moved this information to the Discussion, and provided the key aims of the study in the Background section.

3) Line 110: safer??

This reference to safety is now found in the 6th paragraph (16th line) of the Discussion section. We are suggesting that our constructs achieve improved safety profiles compared to the dosage of pharmacological hormone therapy required to achieve similar results with

our cHRT constructs that delivered physiological levels of hormones. Unfortunately, the higher pHRT doses also caused unwanted side effects such as uterine hypertrophy.

4) Line 146-150: What is the rationale to give 50µg E2 which led to supraphysiological levels? Low doses seem to be effective.

We have clarified that the low dose pHRT treatments were not effective, as evidenced by the FSH levels (Figure 3A) body fat composition (Figure 4C), bone mineral density and bone architecture (Figure 5), or uterine weight and appearance (Figure 6). The low dose pHRT we used corresponded with plasma levels of estrogen and progesterone delivered from our constructs. We then selected the high-dose based on previous studies, and this led to detrimental effects (uterine weight, morphology shown in Figure 6) caused by supraphysiological levels of hormone delivery. These supraphysiological levels of hormones were, however, required to achieve some of the beneficial effects provided by our constructs that secreted lower levels (see Figure 5 on bone).

5) Data concerning bone density and parity, although reassuring, cannot be extrapolated to human.

We do not wish to over-interpret our results in regards to future benefits to humans and we have added a concluding paragraph on the limitations of our studies. However, we believe that our findings are an important first step in developing alternatives to pharmacological strategies of hormone therapy based on the principles of cell therapy and tissue engineering.

6) Discussion: much too long. Too many redundant.

We have reduced the length of the Discussion while highlighting the most important points from our study.

7) Methods: sentences line 528: why 344 rats if n=10/group, and 15 groups?

Our statements are correct. There were 15 groups (now 14) with n = 10 per group. The 344 refers to the strain of rats (Fischer 344 rats). We have now used italics for 344 to denote the difference in this revised manuscript and have indicated throughout the document that this is the strain of rats.

8) L536: Why progesterone only? (controls are already included in the study).

Because an estrogen-only control was used, we felt that a progesterone-only control would also be an appropriate control group for the pHRT in which both hormones were combined.

Reviewer #2 (Remarks to the Author):

The manuscript by Sittadjody et al. describes an in vivo strategy for the construction of a 3-dimensional ovary with granulosa cells and thecal cells for delivery of ovarian hormones. The investigators use rat as the model and place the bioengineered ovary (cell encapsulated ovarian cells) into the omentum. Circulatory steroid hormone levels were maintained at a level intermediate between intact and castrate females (Figure 2) and secondary beneficial effects of

normalized hormone levels were observed (Figure 4, 5).

Overall, this is a nice story that advances the field and may have important long-term consequences on the health of aging women.

Minor point:

1. “Thecal” is an adjective; “theca” is a noun. The authors should correct to be “thecal cells” throughout.

We certainly thank the reviewer for the very complimentary comments to our manuscript. We agree with the reviewer that “theca” is a noun and “thecal” is an adjective. However, to describe this particular ovarian cell type, “theca” is the established name used in the literature and so we have used it in our manuscript in agreement with the established literature.

Reviewer #3 (Remarks to the Author):

Reviewer:

The authors report a new strategy to overcome the problems of pharmacological hormone replacement therapies (pHRT) where high hormones blood levels in short periods of time may lead to adverse effects. This new strategy consists in a cell-based hormone therapy (cHTR) where hormones producing cells are embedded within a biomaterial forming capsules and subsequently, these capsules are implanted in ovariectomized rats. This approach, allow a strict hormone secretion control because the therapeutic cells are stimulated by physiological signals promoting the necessary hormone secretion minute to minute. The manuscript is -in my opinion- original, well written and complete, but some aspects are to be clarified:

1) The mechanical properties of ionically crosslinked alginate capsules can be modified (even coming to break down) due to the diffusion of Ca^{2+} ions.

Did the authors studied the stability and degree of degradation of the constructs after the experimentation process?

We also thank this reviewer for the kind remarks on our manuscript. We have now noted in the revised manuscript that we did not assess the degree of degradation in the constructs (Results section on “Construct implantation, retrieval, and evaluation, last 3 lines).

However, we have noted that the constructs remained intact over the 90 day *in vivo* study as determined by retrieval at the terminal time point (see Figure 1 D-G). We have observed similar behavior *in vivo* and the senior author (Opara) has observed stability of alginate constructs *in vivo* over the course of 1 year.

2) Did the authors test possible effects of the use of calcium on the release of hormones by the encapsulated cells?

We did not assess the effects of the use of calcium on the release of hormones by the encapsulated cells. However, given the similarities of hormone release for non-estrogen hormones with calcium or strontium crosslinked constructs, we believe that this result indicates that there is minimal effect of calcium-dependence. We would also note that sex steroid hormone secretion is not calcium-dependent.

Minor comments:

1) In figure 1G, I,J,K,L,M the scale bar should be bigger.

We have increased the size of the scale bar in our revised Figure 1 for clarity, as suggested.

Referee « Development of an improved bio-mimetic ovarian cell construct for controlled hormone delivery and correction of abnormalities associated with ovarian failure »

The authors have answered our comments and have substantially changed their manuscript accordingly. Some minor concerns remain.

1) Abstract

1. “more effective” can not be extrapolated to human. The conclusion could be confusing for the reader.
2. Relevant low-dose in animal model could be different from relevant low-dose in humans.
3. Sustained stable: please the duration of hormones secretion. The term “sustained stable” is not correct if the duration is not mentioned.
- 2) At the end of the abstract, at least, it should be mentioned isogenic cells...
- 3) P10: Again, this reviewer would like to receive an explanation on the use of a very high dose of E₂ (50µg) which obviously provokes “hyperplasia” and possibly tumor growth. Why not also an intermediate dose? Please, add a sentence. At least, an intermediate dose should be used for further studies.
- 4) Ref 35-36: very old references. The authors should refer to recent literature to explain the “concerns” noted in humans patients on estrogen therapy.
- 5) The reviewer agree with the last sentence of the first paragraph (p10) “challenges of balancing safety and efficacy in HRT treatments”. The authors should add: “taking into account that the results obtained in an animal model can not be extrapolated to human beings (endometrium and breast tissue response so estrogens).
- 6) Discussion: 1st paragraph line 16 “the approaches are not suitable...” Why not? In fact, they are and it should be clearly mentioned.
- 7) P13 Line 17: “potentially suitable” This is an over interpretation. An other proof of concept study with allogeneic or xenogeneic cells is mandatory.
- 8) P13 Line 19: add in animals models
- 9) Last main concern: As the sustained hormone release has a relatively “limited” duration, this reviewer would like to know how the authors will proceed to increase the secretion duration.

Conclusion: Some minor concerns remain but this reviewer would like to congratulate the authors for this “proof of concept” study.

Response to Additional Concerns of Reviewer:

We thank the reviewer very much for congratulating us on the proof-of-concept study described in our manuscript, now entitled “In vivo transplantation of 3D encapsulated ovarian constructs in rats corrects abnormalities of ovarian failure.” We have further revised the manuscript to address the additional concerns raised by the reviewer. Below we address point-by-point each of the comments of the reviewer. Each reviewer comment is provided and our response is indicated in bold type. We have indicated significant changes within the document by yellow highlighting.

Reviewer Comments and Responses

1) Abstract

Comment #1: “more effective” cannot be extrapolated to human. The conclusion could be confusing for the reader.

We have modified the said sentence in the conclusion to specify that the constructs are effective in the rat model to avoid confusion.

Comment #2: "Relevant" low-dose in animal model could be different from "relevant" low-dose in humans.

The modification of that sentence has eliminated the reference to “relevant low-dose”.

Comment#3: Sustained stable: please add the duration of hormones secretion. The term “sustained stable” is not correct if the duration is not mentioned.

We agree with the reviewer and have added the 90-day duration of study to the said sentence.

Comment #4: At the end of the abstract, at least, it should be mentioned isogenic cells.

We have added isogenic cells to the last sentence in the Abstract.

2) P10: Again, this reviewer would like to receive an explanation on the use of a very high dose of E₂ (50µg) which obviously provokes “hyperplasia” and possibly tumor growth. Why not also an intermediate dose? Please, add a sentence. At least, an intermediate dose should be used for further studies.

We have added an explanation for the use of a very high dose of E₂ (50 µg) with relevant references as shown in yellow highlights in the penultimate paragraph of page 14. We have also addressed the issue of an intermediate dose at the bottom of first paragraph on page 10.

3) Ref 35-36: very old references. The authors should refer to recent literature to explain the “concerns” noted in human patients on estrogen therapy.

The two references in question (now #s 29 and 30) were published in 2012. Unfortunately, we are not able to include additional references since indeed we have had to delete many references from manuscript in order to meet the requirements of the journal. Furthermore, we actually have more recent references (#s 3 and 4) in the manuscript that are focused on the subject matter raised by the reviewer.

4) The reviewer agree with the last sentence of the first paragraph (p10) “challenges of balancing safety and efficacy in HRT treatments”. The authors should add: “taking into account that the results obtained in an animal model cannot be extrapolated to human beings (endometrium and breast tissue response to estrogens).

We have modified this sentence (now shown at the bottom of first paragraph on page 10) to address the reviewer’s concern.

5) Discussion: 1st paragraph line 16 “the approaches are not suitable...” Why not? In fact, they are and it should be clearly mentioned.

While we respectfully feel differently and believe that for a postmenopausal woman beyond child-bearing age, a cell-based hormone replacement-only regimen would be a better alternative than a whole egg-containing follicle approach we have modified the sentence to take into consideration the reviewer’s comment, as shown on page 12 (at the end of first paragraph).

6) P13 Line 17: “potentially suitable” This is an over interpretation. Another proof of concept study with allogeneic or xenogeneic cells is mandatory.

We have modified the sentence in question (now at the bottom of second paragraph on page 13 to address the reviewer’s concern.

7) P13 Line 19: add in animals models

We have made this change, as shown in the second paragraph of page 13.

8) Last main concern: As the sustained hormone release has a relatively “limited” duration, this reviewer would like to know how the authors will proceed to increase the secretion duration.

We have addressed this concern as shown in the last paragraph of page 16 that spans between pages 16 and 17.